# The Effects of Botulinum Toxin A Injections on Patients with Radiogenic Lower Urinary Tract Symptoms

**DOI:** 10.3390/toxins17040200

**Published:** 2025-04-15

**Authors:** Anke K. Jaekel, Ann-Christin Brüggemann, John Bitter, Franziska Knappe, Ruth Kirschner-Hermanns, Stephanie C. Knüpfer

**Affiliations:** 1Department of Neuro-Urology, Clinic for Urology, University Hospital Bonn, 53127 Bonn, Germany; 2Department of Neuro-Urology, Johanniter Neurological Rehabilitation Center Godeshoehe GmbH, 53177 Bonn, Germany; 3Clinic for Urology and Pediatric Urology, University Hospital Schleswig-Holstein, Campus Kiel, 24105 Kiel, Germany

**Keywords:** bladder, Botulinum toxin A, radiogenic cystitis, LUTS, quality of life

## Abstract

Botulinum toxin A (BTX-A) injection into the detrusor vesicae is an established therapy for neurogenic lower urinary tract dysfunction as well as idiopathic overactive bladder. Pelvic radiotherapy causes comparable lower urinary tract symptoms (LUTS) in a third of radiated patients. Little is known about the effects of BTX-A injections into the detrusor vesicae in the management of radiogenic LUTS. Our aim was to assess the effect of BTX A injections on these symptoms and related quality of life. **Material and Methods**: In total, 28 patients with BTX-A injections for radiogenic LUTS were assessed retrospectively. We analyzed symptoms recorded in bladder diaries, the results of quality-of-life questionnaires (ICIQ-LUTSqol), and urodynamic studies (UDS) before and after BTX-A injections. **Results**: A significant reduction in daily micturition frequency, nocturia, and pad consumption was demonstrated in the overall cohort and in gender-related subgroup analysis. There was a significant decrease in the ICIQ-LUTSqol independent of gender or BTX-A units. For UDS maximum cystometric bladder capacity (188.0 vs. 258.2 mL, *p* = 0.043), micturition volume (138.2 vs. 216.7 mL, *p* = 0.018), and first desire to void (98.2 vs. 171.2 mL, *p* = 0.042) was significantly improved. No side effects of the toxin injection or urinary retention were observed. **Conclusions**: Intradetrusor injection therapy with BTX-A could represent a safe and effective therapeutic option for radiogenic LUTS with increasing quality of life, reductions in symptoms, and the improvement of urodynamic parameters.

## 1. Introduction

Radiotherapy is one of the most effective ways of treating cancer. Up to 50% of cancer is treated by radiotherapy [1]. In the pelvic area, radiotherapy is used to treat malignancies of the cervix, prostate, rectum, anal canal, vulva, and bladder [1]. Radiogenic complications occur in approximately 5% of cases, with the bladder being particularly susceptible. The incidence of radiogenic cystitis ranges from 23% to 80%. [2]. As the extent of the damage depends on the radiation dose, the fractionation, the type of radiation, and the size of the irradiated portion of the bladder [1,3], there is significant variability in its clinical appearance [4]. The pathogenesis of radiogenic cystitis has not been conclusively clarified down to the molecular level [1]. Basically, the urothelium and the vascular endothelium are damaged by ionizing radiation [3]. The consequences are inflammatory reactions and impairment of the urothelial barrier function to protect the underlying tissue from noxious substances in the urine [3,5]. Fibrosis and the loss of bladder compliance can occur further down the line of treatment [3,5]. Besides pain and macrohematuria, radiogenic cystitis can result in symptoms of the lower urinary tract (LUTS) such as urgency, pollakiuria, urinary incontinence, and frequency [6]. For those affected, radiogenic cystitis is a major burden with a severe reduction in quality of life [6], including lack of sleep and social isolation [5].

A variety of therapeutic approaches exist, such as medication, immunomodulatory/suppressive therapies, instillations with formalin or aluminum, hyperbaric oxygenation, or laser therapy [1]. These approaches are sometimes associated with side effects and are time-consuming or only effective in some aspects of complex symptoms [6].

Detrusor injection therapy with Botulinum toxin A (BTX-A) is a long-established procedure in the treatment of neurogenic detrusor overactivity [7]. Besides the inhibition of acetylcholine at the neuromuscular junctions with a therapeutic effect on muscular spasticity and over-contraction in neurogenic lower urinary tract dysfunction, other effects could be demonstrated for BTX-A. The release of sensory neuropeptides is also inhibited by the toxin. Furthermore, pain-inhibiting and anti-inflammatory effects have been described via its inhibitory effect on vanilloid receptors and cyclooxygenase-2 [8]. The clinical use of BTX-A was, therefore, extended to various non-neurogenic dysfunctions of the lower urinary tract, e.g., idiopathic overactive bladder (OAB) in adults [9]. Radiogenic cystitis can result in LUTS, which are similar to the symptoms of OAB and neurogenic detrusor overactivity. Therefore, there is a rationale for the therapeutic use of BTX-A in cases of radiogenic cystitis, but the use of BTX-A in this indication has only been investigated in one study with a very small sample size [6]. Urodynamic data are completely lacking to this day. This study aimed to improve data on the effect of BTX-A detrusor injection therapy on LUTS, quality of life, and urodynamic parameters in patients with radiogenic cystitis.

## 2. Results

### 2.1. Intervention-Specific Data

The intervention was performed under general anesthesia in 27 patients (23 undergoing intubation and 4 provided with a laryngeal mask) and under spinal anesthesia in one patient. The mean hospitalization time was 1.85 days (SD ± 1.1, min 2; max 5). The mean time to catheter removal was 1.41 days (SD ± 0.6, min 1; max 3). Diagnosis with ultrasonography 92.9% (26/28) showed no post void residual (PVR) after the injection; two patients with PVR had values of 12 mL and 78 mL, respectively. None of the patients had systemic side effects, macrohematuria, or urinary retention. In total, 35.7% (10/28) had consecutive injections when the effect wore off.

### 2.2. The Effects of BTX-A Injections on LUTS

LUTS decreased statistically significantly in the overall cohort for all three parameters assessed (daily micturition frequency (DMF), nocturia, pads per 24 h (PPD)) in the before and after comparison. These significant reductions were also seen in the subgroups separated by gender (except PPD in men) and separated by the units of BTX-A injected in the 200 U subgroup (except PPD). The BTX-A 100 and 300 U subgroups of our cohort did not show a statistically significant reduction in LUTS. Table 1 provides an overview of the pre-interventional and post-interventional LUTS.

### 2.3. The Effects of BTX-A Injections on Quality of Life

There were 18/28 valid samples for the International Consultation on Incontinence Questionnaire-Lower Urinary Tract Symptoms Quality of Life Module (ICIQ-LUTSqol). The comparison of quality of life before and after the injection of BTX-A showed a significant difference when considering all patients and when differentiating by gender. All analyses showed a strong Cohen’s d effect (Cohen’s d > 0.8) on the results of the ICIQ-LUTSqol, which was lowest in the female subgroup. When analyzed according to the units of BTX-A applied, the higher the total dose, the greater the differences in the mean value of the ICIQ-LUTSqol results before and after the intervention. However, only a few samples were available for 300 U. Table 2 gives a detailed overview of the results regarding the ICIQ-LUTSqol.

### 2.4. The Effects of BTX-A Injections on Urodynamic Outcomes

A limited number of variables (6/28) were available for the evaluation of urodynamic data. In comparison from pre- to post-intervention, all urodynamic parameters indicated an improved bladder storage function, but a statistically significant improvement could only be assumed for maximum cystometric bladder capacity (MCBC), micturition volume (MV), and the first desire to void (FDV) based on the available data. Two patients showed detrusor overactivities in the pre-interventional urodynamic study (UDS), and they remained in the post-interventional UDS. One person had detrusor overactivity in the post-interventional UD due to a larger storage capacity.

There was a significant reduction in PVR from a pre-interventional median of 60 mL to 0 mL post-intervention in the UDS. Table 3 shows the urodynamic results and the comparison pre- and post-intervention. Figure 1 shows a graphical overview of the urodynamic parameters before and after BTX-A injection therapy.

## 3. Discussion

Detrusor injection therapy with BTX-A is a frequently used and safe therapeutic option in urology. It has become an integral part of the treatment of neurogenic detrusor overactivity and idiopathic OAB in recent years [10]. Symptoms of the lower urinary tract in interstitial cystitis or idiopathic OAB are similar to those in radiogenic cystitis [1]. The consequences for those affected by radiogenic cystistis are immense, and long-term effective treatment options are limited [8]. One of these treatment options is the intravesical injection therapy of BTX-A, but there are limited data available for this special indication [8]. Therefore, we retrospectively analyzed our available data on BTX-A therapy in radiogenic cystitis.

In our study, we obtained evidence that the application of the toxin had a significant effect on the improvement of quality of life on some urodynamic parameters and on micturition frequency during the day and at night, as well as on the PPD in some sub-analyses. Interestingly, BTX-A in our cohort had a positive effect on some aspects of LUTS, QoL, and urodynamics, though not on others. These aspects will be addressed in the following discussion on the basis of the available literature.

The LUTS improved in the before/after comparison in the group for all BTX-A units, in the subgroups of both women and men, and in the 200 U subgroup. Although the latter two showed a reduction in PPD, no statistical significance could be demonstrated for these LUTSs. This may be because the outcome of PPD is highly variable. Different pad types and qualities can absorb different amounts of urine [11]. How much urine has been collected remains unclear if only the number of pads is recorded. Measurements of urine loss using weighed pads would provide more accurate values [12], but this method is not routinely used over several days due to the increased effort involved. However, it would be useful for further studies to precisely record the amount of involuntary urine loss and, thus, to make more reliable statements.

In the subgroup analysis of the total dose, the use of 200 U in contrast to 100 and 300 U showed significant results in the reduction in LUTS. This effect may be due to the limited sample size [13] and would, therefore, need to be investigated in larger collectives, ideally in a prospective setting. In a similar indication, Vuong et al. in 2011 prospectively demonstrated a statistically significant reduction in urgency and bowel frequency in 15 people with acute radiation proctitis after the application of 100 U of BTX-A [14]. As the only study on the use of BTX-A in people with radiogenic cystitis, five out of six cases reported an improvement in the symptoms recorded by the bladder diary after the application of 200 U in a retrospective setting [8]. However, the statistical analysis in this study was only descriptive, so we could not draw any more detailed parallels with our data. The use of 100 or 300 U in the indication of radiogenic cystitis has not yet been described in the literature to date. In a prospective study comparing the number of urinary incontinence episodes experienced in groups receiving 200 U, 300 U, or a placebo with spinal cord syndrome and multiple sclerosis, both BTX-A doses showed a significant reduction in urinary incontinence compared to the placebo. According to this, there was no significant difference between the two doses [15]. The discrepancy observed in our results can be attributed to the distinct etiology of bladder dysfunction and the utilization of a primary outcome measure that encompasses neurogenic urinary incontinence alone. The predominantly inflammatory vascular damage of radiogenic cystitis may react differently to the toxin compared to the neurogenic dysfunction of the lower urinary tract. Furthermore, our study was not designed to show differences between the two doses. We performed a longitudinal comparison of related samples.

The effect of BTX-A injection on LUTS-related QoL (quality of life) showed a significant increase in QoL in the overall cohort and the subgroup analyses by the number of units and gender. This positive effect has already been described several times in other urological indications for BTX-A applications. Giannantoni et al. were able to show similar results in a prospective study of 14 women suffering from bladder pain syndrome. Three months after the application of 200 U of BTX-A, the quality of life of the participants was significantly improved, and anxiety and depression were significantly reduced. Bladder capacity, urgency, and frequency were improved [16]. Botulinum toxin also showed positive effects on quality of life and urodynamic parameters in people with neurogenic symptoms of the lower urinary tract [15,17]. In our cohort, the positive effect on quality of life was less distinct in women than in men, although the reduction in pad use was statistically more distinct in women. This could be because women feel less impaired in their QoL by wearing pads and being incontinent than men [18]. This was also shown for fecal incontinence [19]. The lower pre-interventional mean value of the ICIQ-LUTSqol of women compared to men supports this thesis, as the QoL of women appears to be less impaired in advance. LUTS and urinary incontinence appear to have a different impact on the quality of life for men. Studies show that the impact of urinary incontinence on their ability to work and have a sex life is greater [18,20].

The sub-analysis separated into the BTX-A units showed that the more BTX-A used, the greater the increase in QoL. However, a reliable assessment cannot be derived from the very small amount of data in the subgroup analysis, even if the increase in QoL is statistically significant in all subgroups and the overall analysis. The dose-dependent effects of BTX-A on health-related QoL were also investigated by other authors. The comparison of 200 to 300 U in neurogenic LUTSs showed no differences between the two doses in health-related QoL [15]. In 2010, Dmochowski et al. examined the dose-dependent increase in QoL in patients with idiopathic OAB, starting with 50 U. Initially, there was an increase in quality of life with increasing BTX-A doses, which then stagnated at 100 U [9]. Therefore, both studies showed similar results for the 200 and 300 U dose ranges. Both studies also showed an optimum dose of BTX-A for the respective indication, at which the intended effect and side effects were balanced [9,15]. For the present indication of radiogenic cystitis, such an optimum dose could also exist. A sophisticated study protocol would be needed to find this optimum dose. This protocol could then be used to compare the effects of the different doses in terms of efficacy, health-related QoL, and side effects to answer the currently unresolved aspects.

In our work, we revealed hints that the effect of BTX-A in radiogenic cystitis is also detectable in UDS. At this point, once again, we would like to emphasize the exploratory, retrospective nature of our study. Only a few complete data sets on urodynamics before and after the application of BTX-A were available. The urodynamic parameters of MCBC, FDV, MV, and PVR showed statistically significant changes. A systematic review by Mangera et al. also reported an improvement in MCBC for neurogenic detrusor overactivity and idiopathic OAB after BTX-A applications. For other causes of LUTS, the data situation was so heterogeneous that no statements could be made on urodynamic results [21]. According to the current guidelines, studies since 2014 have investigated urodynamic outcomes after BTX-A injection therapy for causes of LUTS other than neurogenic detrusor overactivity or idiopathic OAB, which are rare [22,23]. In the work by Kuo and Kuo 2016, the BTX-A effect of 100 U for idiopathic OAB was compared with interstitial cystitis/bladder pain syndrome (IC/BPS) with a focus on adverse events [24]. The parameters of bladder capacity, Qmax, PVR, and voiding efficiency (the proportion of the voided volume of the total bladder capacity) were determined at baseline and over the course of the study. The OAB group showed a significant increase in bladder capacity and residual urine and a decrease in voiding efficiency. The IC/BPS patients did not show these significant changes. The significant change in the IC/BPS group was due to the increase in Qmax after 6 months. Comparable to our results, there was a non-significant increase in Qmax after BTX-A from baseline over the course of the study. A limiting factor in this study was that Qmax was measured at baseline by UDS and over time by uroflowmetry. There was no measuring catheter in the urethra during the follow-up checks; this situation could explain the improved Qmax. On the other hand, a reduction in urgency (a reduced FDV in our cohort) could lead to a more direct and physiological micturition, as the sphincter and detrusor function is more synergistic. To our knowledge, there are no studies on this fact. However, in reverse, van Brummen et al. in 2004 showed that urgency is associated with reduced MV and increased bladder sensitivity [25]. The rationale of the study by Kuo et al. was the different effects of BTX-A on urodynamic parameters for different indications [24]. This rationale and the theory of directed micturition could also explain the reduction in PVR after BTX-A in our cohort. It remains unclear in our study setting if the PVR result is due to an outlier in the pre-interventional PVR measurement.

Our results raise further questions regarding the detailed effect of BTX-A on symptoms of radiogenic cystitis. To clarify these questions, prospective, randomized, placebo-controlled studies utilizing bladder diaries, weighed pads, and UDS should be conducted. This will help to clarify the value of BTX-A for the treatment of radiogenic LUTS.

## 4. Conclusions

This study aimed to assess the effect of BTX-A in patients with radiogenic LUTS regarding QoL, symptom control, and urodynamic parameters. We found a significant reduction in frequency, nocturia, pad consumption, and improvements in LUTS-related QoL and urodynamic parameters. No adverse events were documented. This indicates that intradetrusor injection therapy with BTX-A could represent a safe, therapeutic option for radiogenic LUTS. Prospective placebo-controlled trials must confirm these findings.

## 5. Materials and Methods

This retrospective study included 28 patients who had been treated with BTX-A injections between 09/2008 and 01/2019 in the clinic of urology of a university hospital. Inclusion criteria included having received radiotherapy as part of oncological treatment in the pelvic region and the presence of LUTS. We excluded patients with neurodegenerative diseases, paraplegia/tetraplegia, other neurological conditions, and pre-existing urinary incontinence for reasons other than radiogenic damage. We captured LUTS as daily micturition frequencies (DMF), the number of nocturia episodes, and the number of incontinence pads used per 24 h (PPD) from a 24 h bladder diary. The data were based on an observation period of up to 6 months before the injection and at least 3 weeks after the BTX-A injection. All interventional procedures were conducted under general or regional anesthesia.

The study population consisted of 64% (18/28) males and 36% (10/28) females. The average age for the entire population was 74.5 (SD ± 8.1, min 56; max 90) years. The average age of the men was 74.2 (SD ± 7.2, min 61; max 90), and that of the women was 75.1 (SD ± 9.9, min 56; max 90) years. In 46% (13/28), 100 units (U) were used; in 36% (10/28), 200 U were used; and in 18% (5/28), 300 U of BTX-A was administered. All patients were treated with onabotulinumtoxinA. OnabotulinumtoxinA was reconstituted and diluted in sterile, non-preserved 0.9% saline. The total dose and the number of injection sites were determined according to the surgeon’s preference. The injections were made into the detrusor muscle, distributed across the lateral and posterior bladder walls, avoiding the trigonum vesicae. Table 4 provides an overview of the number of patients treated, categorized according to dose, injection sites, and units per mL.

Urodynamic studies were performed as video-urodynamics according to the current doctrine [26]. The following individual urodynamic parameters were evaluated: first desire to void in mL, strong desire to void in ml, maximum cystometric bladder capacity in mL, compliance in mL/cm H_2_O, micturition volume in mL, maximum urinary flow in mL/s, and maximum detrusor pressure in the micturition phase in cm H_2_O. For the measurement of the urinary symptoms related to quality of life, the International Consultation on Incontinence Questionnaire-Lower Urinary Tract Symptoms Quality of Life Module in the German version was used [27]. This questionnaire is designed to record the effects of LUTS on quality of life as well as to evaluate treatment modalities. The questionnaire focuses primarily on social activities in everyday life and consists of 20 questions relating to the past 4 weeks. Sixteen of these questions are scored from 1 (not at all, never) to 4 (affects me a lot, always). A score of between 19 and a maximum of 76 points can be achieved. The higher the score, the greater the impairment of quality of life.

The statistical analysis was performed using SPSS^®^, version 29.0 (IBM Corp., Armonk, NY, USA). Histograms and the Kolmogorov–Smirnov test were used to test for normal distribution. For data analysis, the non-parametric Wilcoxon test was used for related, non-normally distributed variables. The paired *t*-test was used for normally distributed variables. Cohen’s d was determined to calculate the effect size for the *t*-test. A significance level < 0.05 was considered statistically significant.

The study was conducted in accordance with the Declaration of Helsinki and approved by the Ethics Committee of the Medical Faculty of the Christian-Albrechts-University Kiel, Germany on 20 November 2017 under the number D 559/17.

## 6. Limitations

The study has several limitations, which must be addressed as follows: The main limitation of our study was its retrospective nature and the associated small number of valid samples in the small cohort. However, all data from a university hospital with a functional urological focus were evaluated over a period of more than 10 years. Individuals with pre-existing neurological conditions or urinary incontinence were excluded from the study, which further reduced the number of patients. In the literature, no larger case numbers on this topic have been described over a longer period of time [6], so the evaluation of these data nevertheless seems to be useful. Another important point is the selection of the doses of BTX-A used and the number of the sites applied, which were subject to the surgeon’s preference. On the other hand, even in on-licensed applications of BTX-A, there was a wide range of injection sites and dosages used [28]. It must be emphasized that, to date, this injection is used off-label without any specifications or systematic data in the literature regarding its application in radiogenic cystitis. This again emphasizes the necessity of systematic, prospective studies to assess the potential of BTX-A for radiogenic cystitis. Furthermore, we have not evaluated the information on the primary tumor, the time of application, or the time interval between radiation and BTX-A injection. The impact of these factors and the effect of BTX-A on radiogenic cystitis must also be addressed in further investigations.

## Figures and Tables

**Figure 1 toxins-17-00200-f001:**
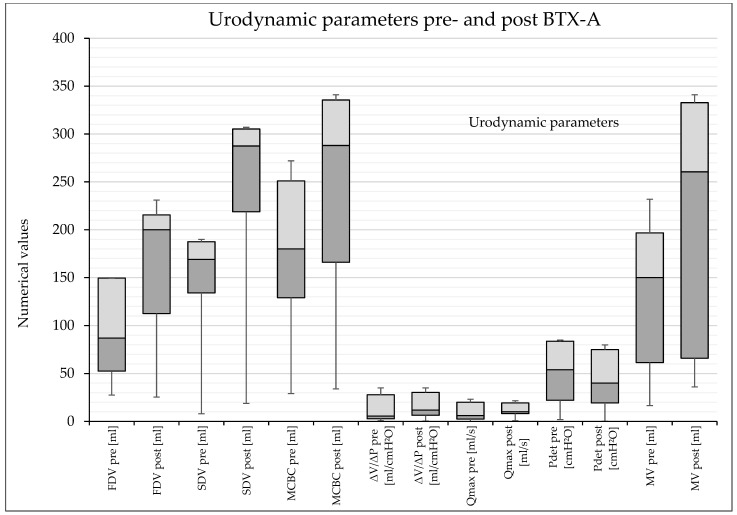
Boxplots of the urodynamic parameters before and after BTX-A therapy.

**Table 1 toxins-17-00200-t001:** Overview of the descriptive statistics and the pre- and post-intervention comparison of lower urinary tract symptoms.

		Pre-Intervention			Post-Intervention			
	LUTS [Number]	N/28	Median (IQR)	Mean (SD, Min; Max)	N/28	Median (IQR)	Mean (SD, Min; Max)	Wilcoxon Test Z (*p*)
total	DMF	26	10.0 (7.75–12.00)	10.85 (±5.1, 4; 24)	26	5.50 (4.0–7.25)	7.04 (±5.6, 1; 24)	−3.24 (0.001)
	Nocturia	22	4.0 (3.0–4.25)	3.82 (±1.7, 0; 8)	22	2.5 (1.0–3.0)	2.36 (±1.1, 1; 4)	−3.16 (0.02)
	PPD	23	2.0 (1.0–5.0)	2.78 (±2.6, 0; 8)	23	1.0 (0.0–4.0)	1.83 (±1.9, 0;5)	−2.07 (0.039)
Male	DMF	16	10.0 (7.25–11.5)	9.38 (±2.8, 4; 15)	16	5.5 (3.25–7.75)	6.44 (±5.2, 1; 24)	−2.31 (0.021)
	Nocturia	14	4.0 (2.75–4.0)	3.71 (±1.32, 2; 7)	14	3.0 (1.75–3.0)	2.5 (±2.6, 1; 4)	−2.52 (0.012)
	PPD	15	1.0 (1.0–3.0)	2.33 (±2.5, 0; 8)	15	1.0 (0.0–3.0)	1.6 (±1.6, 0; 5)	−0.96 (0.337)
Female	DMF	10	11.1 (7.5–21.0)	13.2 (±7.2, 5; 24)	10	5.5 (4.0–9.0)	8.0 (±6.5, 4; 24)	−2.37 (0.018)
	Nocturia	8	4.0 (3.0–5.0)	4.0 (±2.27, 0; 8)	8	2.0 (1.0–3.0)	2.13 (±1.23, 1; 4)	−2.01 (0.044)
	PPD	8	4.5 (0.5–5.75)	3.62 (±2.67, 0; 7)	8	2.0 (0.0–4.75)	2.25 (±2.24, 0; 5)	−2.06 (0.039)
BTX-A 200 U	DMF	8	10.0 (10.0–12.00)	11.13 (±1.81, 10; 15)	8	5.5 (3.25–8.00)	5.5 (±2.78, 1; 9)	−2.53 (0.012)
	Nocturia	7	4.0 (3.0–4.0)	4.0 (±1.53, 2; 7)	7	3.0 (1.0–3.0)	2.43 (±1.13, 1; 4)	−2.26 (0.024)
	PPD	8	1.5 (0.25–3.0)	2.25 (±2.61, 0; 8)	8	0.5 (0.0–2.0)	0.88 (±0.99, 0; 2)	−1.28 (0.202)

LUTS—lower urinary tract symptoms; DMF—daily micturition frequency; PPD—incontinence pads per 24 h. N—number of valid values; BTX-A—botulinum toxin A.

**Table 2 toxins-17-00200-t002:** Overview of the comparison of the pre- and post-interventional International Consultation on Incontinence Questionnaire-Lower Urinary Tract Symptoms Quality of Life Module ICIQ-LUTSqol by subgroup.

ICIQ-LUTSqol								
Subgroup	N/28	Mean Pre SD	Mean Post SD	Mean Diff SD	Confidence Interval	T	*p * (One-Sided)	Cohen’s d
overall	18	52.78 ± 8.52	40.28 ± 13.93	12.50 ± 12.05	6.51–18.49	4.403	0.000	1.038
male	10	56.40 ± 8.63	40.30 ± 15.96	16.1 ± 13.45	6.48–25.72	3.784	0.002	1.197
female	8	48.25 ± 6.21	40.25 ± 12.0	8.0 ± 8.83	0.62–15.39	2.562	0.019	0.906
BTX-A 100 U	9	53.67 ± 9.45	48.11 ± 9.29	5.56 ± 4.42	2.16–8.95	3.772	0.003	1.257
BTX-A 200 U	5	51.40 ± 7.80	36.00 ± 15.10	15.4 ± 10.11	2.84–27.96	3.405	0.014	1.523
BTX-A 300 U	4	52.50 ± 9.33	28.00 ± 12.25	24.5 ± 16.82	2.27–51.27	2.913	0.031	1.546

BTX-A—Botulinum toxin A; N—number of valid values; mean pre—pre-intervention; mean post—post-intervention; mean diff—the difference between pre- and post-interventional consultation.

**Table 3 toxins-17-00200-t003:** Overview of the descriptive statistics and comparison of the urodynamic parameters and residual urine before and after the intervention.

	Pre-Intervention			Post-Intervention			
Parameters	N/28	Mean (SD, Min; Max)	Median (IQR)	N/28	Mean (SD, Min; Max)	Median (IQR)	Wilcoxon Test Z (*p*)
FDV [mL]	5	98.20 (±52.63, 25; 150)	87.00 (52.5–149.5)	5	171.20 (±57.94, 87; 231)	200 (112.5–215.5)	−2.032 (0.042)
SDV [mL]	4	163.50 (±28.35, 126; 190)	169.0 (134.0–187.5)	4	270.50 (±48.97, 200–307)	287.5 (218.75–305.25)	−1.826 (0.068)
MCBC [mL]	5	188.00 (±66.2, 100; 271)	180.0 (129–251.0)	5	258.20 (±89.74, 132; 341)	288.00 (166–335.5)	−2.023 (0.043)
Compl [mL/cmH_2_O]	4	12.00 (±15.43, 2.0; 35)	5.5 (2.75–27.75)	4	16.13 (±13.35, 6.0; 35)	11.75 (6.35–30.28)	−1.000 (0.317)
Qmax [mL/s]	5	10.20 (±9.35, 2.4–23.2)	6.1 (2.45–20.0)	5	12.96 (±6.05, 7.4; 21.6)	10.0 (8.15–19.25)	−0.674 (0.500)
Pdet [cmH_2_O]	4	53.25 (±33.99, 20; 85)	54.00 (22.00–83.75)	4	44.75 (±30.28, 19; 80)	40.00 (19.25–75.00)	−1.241 (0.180)
MV [mL]	6	138.17(±71.86, 45; 232)	150.0 (61.5–196.75)	6	216.67 (±132.46, 30; 341)	260.5 (66.0–332.75)	−2.37 (0.018)
PVR [mL]	7	56.57 (±31.31, 0; 100)	60.0 (45.0–81.0)	7	19.57 (±24.59, 0; 50)	0.0 (0–47.0)	−2.21 (0.027)

FDV—first desire to void; SDV—strong desire to void; MCBC—maximum cystometric bladder capacity; Compl—bladder compliance; Qmax— maximum urinary flow; MV—micturition volume; PVR—post void residual.

**Table 4 toxins-17-00200-t004:** Detailed overview of the BTX-A doses applied.

Total Dose BTX-A	N/28	Number of Injection Sites	Number of Patients	Dilution/ Preparation (Units per mL)	mL per Injection Site
100	13	7	1/13	10	nd
		10	6/13	10	1.0
		17	4/13	6.7	1.0 ^#^
		20	2/13	10	0.5
200	10	10	1/10	20	1.0
		17	1/10	13.3	1.0 ^#^
		20	5/10	10	0.5
		21	2/10	10	0.5 ^#^
		28	1/10	6.7	nd
300	5	10	1/5	30	1.0
		17	1/5	20	1.0 ^#^
		20	1/5	15	1.0
		30	1/5	10	1.0
		50	1/5	10	0.5/1.0

nd: The exact amount per injection could not be determined because the prescribed application was deviated intraoperatively due to intravesical bleeding. ^#^ The total dose was dissolved in 15 mL, with an additional injection of 0.9% saline before and after the injections to flush the needle.

## Data Availability

The original contributions presented in this study are included in the article. Further inquiries can be directed to the corresponding authors.

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
