# Peer review of "The Effects of Botulinum Toxin A Injections on Patients with Radiogenic Lower Urinary Tract Symptoms"

_toxins, 2025, doi:10.3390/toxins17040200_

Round 1
Reviewer 1 Report
Comments and Suggestions for Authors
The authors have addressed an important topic concerning the effect of BoNT/A on the relief of LUST symptoms. The article contains significant retrospective observations, on a relatively small number of patients. An important element of the work is the urodynamic changes before and after botulinum toxin injection, which in my opinion should be described in more detail. The main weaknesses of the work are the lack of description of the methods of injection of patients, the type of preparations that were administered to patients. Different preparations of botulinum toxins differ in activity, so this information is very important from the point of view of assessing effectiveness. Doses are given, but they do not refer to specific types of BTX-A (e.g. Abo; Inco; Ona; etc.). Very important information, especially with a small number of patients, are the methods of injection, the places where botulinum toxins were applied. The number of groups of patients receiving different doses of unknown botulinum toxin preparations differ significantly (100U - 13 patients, and 300U 5 patients). No far-reaching conclusions can be drawn on this basis. Data on the patient population should be moved from the "Results" section to "Material and methods". I recommend supplementing the above-mentioned information. I also recommend preparing the manuscript rather in the form of "communication". The presented data are important, but further research is needed, verification on a larger group of patients, a comprehensive presentation of the criteria for qualifying patients for botulinum toxin injections, reasons for selecting specific injection doses. If these data are partially supplemented, then in my opinion the data can be published.
Reviewer 2 Report
Comments and Suggestions for Authors
Dear authors,
I read your paper with interest, considering the topic. First of all, I have troubles reading it until discover that there was a Material and Methods section at the end of the study. This should be moved up before the results, as always.
In methods an internal review board is needed. Also is not clear how you decided the dose of Botulinium for each patients. It’s also not clear the primary tumor of these patients: the reason for this comment is that, in my opinion, a bladder cancer treated with RT received more radiation than a colon’s tumor, resulting in different symptoms and a different microscopic anatomy of the bladder detrusor.
The procedure should be described in methods.
Why IPSS was not assessed in male patients?
Abstract: “percutaneous radiation therapy” is it right to call it percutaneous?
Results:
- showed increased bleeding intraoperatively: this can happen during an endoscopy, it should be reported? Does it add something? I think it can be removed.
- DMF: how do you collect this?
However, looking forward to see your comments because the idea is good and the topic appealing.
Reviewer 3 Report
Comments and Suggestions for Authors
In this article the authors, aimed to demonstrate a beneficial effect of Botulin toxin A (BTX-A) on lower urinary tract symptoms in patients that have undergone radiotherapy. The study is valid and presents some novelty. It contribute to enrich the literature. Nevertheless, some issues need to be fixed to achieve the sufficient priority for publication.
The introduction effectively introduces the topic and the study’s importance, but it would benefit from improved clarity, more detailed mechanistic insights, and a stronger justification for BTX-A as a therapeutic approach. Addressing these points would enhance the manuscript's impact and readability. Some sentences are overly complex or fragmented, affecting readability. For instance, the sentence:
"Radiation-induced complications occur in around 5% of cases. The bladder is one of the organs particularly affected by radiation damage with an incidence of radiogenic cystitis of 23-80%."
could be revised for smoother flow:
"Radiation-induced complications occur in approximately 5% of cases, with the bladder being particularly susceptible. The incidence of radiogenic cystitis ranges from 23% to 80%, depending on various factors." The manuscript uses both "radiogenic cystitis" and "radiation-induced cystitis" interchangeably. Standardizing the terminology throughout the text would improve clarity and consistency. While the introduction states that "the pathogenesis of radiogenic cystitis has not been conclusively clarified at the molecular level", it does not briefly summarize the existing molecular insights. Including more detail on oxidative stress, inflammatory cytokines, or vascular changes involved in radiation-induced bladder damage would strengthen the mechanistic understanding. The introduction presents multiple treatment options but does not explain why BTX-A might be particularly beneficial for radiogenic cystitis beyond symptom similarity with OAB. Briefly discussing its mechanism of action (e.g., inhibition of acetylcholine release and modulation of sensory pathways) could provide a stronger rationale for its use in this context. The final paragraph states that "the use of BTX-A in radiogenic cystitis has only been investigated in one study," but it does not elaborate on the findings of that study. A brief mention of its key limitations (e.g., small sample size, and lack of urodynamic data) would reinforce the need for further research.
The discussion section is well-structured and informative but would benefit from a stronger focus, more precise statistical interpretation, and a clearer central message. Refining these aspects will improve the overall clarity and impact of the manuscript. The discussion covers a wide range of topics, but it lacks a strong central takeaway. A concise summary at the beginning or end of the discussion would help emphasize the most critical findings and their implications.While comparing the study findings to prior research is valuable, some comparisons (e.g., studies on spinal cord injury, multiple sclerosis, and fecal incontinence) may not be directly relevant to radiogenic cystitis. Several sentences are unnecessarily complex or repetitive. For example:
"The difference to our results may be explained by the different causes of bladder dysfunction and the choice of primary outcome, as only neurogenic urinary incontinence was considered here." The discussion briefly mentions significant changes in MCBC, FDV, MV, and PVR, but does not provide a clear pathophysiological explanation for these findings.
Terms such as “LUTS,” “overactive bladder,” and “urinary incontinence” are used interchangeably without clear distinction in some sections.
Comments on the Quality of English Languagesome sentences are overly complex or fragmented, affecting readability. For instance, the sentence:
"Radiation-induced complications occur in around 5% of cases. The bladder is one of the organs particularly affected by radiation damage with an incidence of radiogenic cystitis of 23-80%."
Reviewer 4 Report
Comments and Suggestions for Authors
Main concerns
Too small population studied and insufficient description.
It would be useful to make a paragraph on materials and methods which would allow a better description of the population
There are significant problems in understanding the paragraph « Effects of BTX-A injections on quality of life » with 2 incompatible tables.
Very small populations in paragraph « Overview of the descriptive statistics and comparison of the urodynamic parameters and residual urine before and after the intervention » (table 4).
Minor remarks :
1- about abbreviations H2O in place of H2O ; mL in place of ml
2- references 7 and 8 are the same
Reviewer 5 Report
Comments and Suggestions for Authors
There is no control group.
There is no extensive discussion of side effects.
Some patients had LUTS caused by BPH and the advanced age of some patients which are two major drawbacks - risk of bias.
On what criteria were the different doses chosen?
Round 2
Reviewer 1 Report
Comments and Suggestions for Authors
All my previous comments seem to have been taken into account. The text of the paper is much clearer. I recommend the manuscript for publication.
Reviewer 2 Report
Comments and Suggestions for Authors
Dear Authors, I appreciate the point-by-point reply to my comments. The answer provide sufficient informations to clarified my doubts. No further comments
Reviewer 3 Report
Comments and Suggestions for Authors
thank you for having addressed all my comments and requests.
Reviewer 4 Report
Comments and Suggestions for Authors
The author responded adequately to my comments.
Reviewer 5 Report
Comments and Suggestions for Authors
I am agree with the revisions .However, the ethics committee approval must be clarified.